# Human Activity Recognition Based on Deep Learning and Micro-Doppler Radar Data

**DOI:** 10.3390/s24082530

**Published:** 2024-04-15

**Authors:** Tan-Hsu Tan, Jia-Hong Tian, Alok Kumar Sharma, Shing-Hong Liu, Yung-Fa Huang

**Affiliations:** 1Innovation Frontier Institute of Research for Science and Technology, National Taipei University of Technology, Taipei 10608, Taiwan; thtan@ntut.edu.tw; 2Department of Electrical Engineering, National Taipei University of Technology, Taipei 10608, Taiwan; t109318069@ntut.org.tw; 3Department of Computer Science and Information Engineering, Chaoyang University of Technology, Taichung City 41349, Taiwan; rbaloksharma@gmail.com; 4Department of Information and Communication Engineering, Chaoyang University of Technology, Taichung City 41349, Taiwan; yfahuang@cyut.edu.tw

**Keywords:** deep learning, human activity recognition, radar sensor, micro-Doppler effect, cross-channel operation

## Abstract

Activity recognition is one of the significant technologies accompanying the development of the Internet of Things (IoT). It can help in recording daily life activities or reporting emergencies, thus improving the user’s quality of life and safety, and even easing the workload of caregivers. This study proposes a human activity recognition (HAR) system based on activity data obtained via the micro-Doppler effect, combining a two-stream one-dimensional convolutional neural network (1D-CNN) with a bidirectional gated recurrent unit (BiGRU). Initially, radar sensor data are used to generate information related to time and frequency responses using short-time Fourier transform (STFT). Subsequently, the magnitudes and phase values are calculated and fed into the 1D-CNN and Bi-GRU models to extract spatial and temporal features for subsequent model training and activity recognition. Additionally, we propose a simple cross-channel operation (CCO) to facilitate the exchange of magnitude and phase features between parallel convolutional layers. An open dataset collected through radar, named Rad-HAR, is employed for model training and performance evaluation. Experimental results demonstrate that the proposed 1D-CNN+CCO-BiGRU model demonstrated superior performance, achieving an impressive accuracy rate of 98.2%. This outperformance of existing systems with the radar sensor underscores the proposed model’s potential applicability in real-world scenarios, marking a significant advancement in the field of HAR within the IoT framework.

## 1. Introduction

In recent years, with the declining birth rate and significant improvements in medical standards, the age structure of populations in various countries worldwide has been aging rapidly. Taking Taiwan as an example, data from the National Development Council’s population estimation system reveal that the young and middle-aged population has been decreasing annually since its peak in 2012 [1]. The elderly population surpassed the young population in 2016 and exceeded 14% of the total population in 2018, marking Taiwan’s transition into an aged society. It is projected that Taiwan will become a super-aged society by 2026, with the elderly population surpassing 20% of the total population [1]. When the birth rate continues to fall and the average age increases, the burden on the young and middle-aged population to support the elderly will intensify. Caregivers will need to attend to more elderly individuals, reducing the average time available for caregiving and increasing the risk of the cared-for individuals being left alone.

Human activity recognition (HAR) employs various sensors to collect information on user movement and recognizes the user’s activities based on their unique movement patterns [2]. It provides real-time monitoring and assists in determining whether the user’s daily activities are normal. For instance, for healthy users, daily exercise routines and bedtimes can be set, with reminders issued through Internet of Things (IoT) devices [3]. Additionally, in the event of an unexpected incident, the system can notify family members and medical units to prevent delays in receiving medical treatment. 

Research on HAR primarily utilizes cameras and sensors [2,4]. Depth cameras, utilizing near-infrared radar, and RGB cameras are two camera types [5]. Depth cameras struggle with background interference, while RGB cameras excel in recognition but risk user privacy and light interference. Wearable sensors collect data via inertial measurement units (IMUs) but can be uncomfortable; radar sensors, in contrast, offer privacy and are unaffected by light or obstructions, providing greater versatility [6]. Due to their non-contact nature and ability to protect privacy, radar sensors are increasingly used in fields such as human–computer interaction, smart living, and health management for HAR [7,8]. Traditional machine learning methods, which often rely on manual selection of features such as average values and standard deviations, can overlook distinctive characteristics, leading to lower recognition accuracy [9].

In the previous studies, Wang et al. [10] combined stacked recurrent neural networks (RNNs) with long short-term memory (LSTM) and deep convolutional neural networks (CNNs) to classify six human actions—boxing, clapping, waving, walking in place, jogging, and walking—using radar Doppler images. Results showed the RNNs with LSTM achieved 92.65% accuracy, while deep CNNs reached 82.33%. The study utilized the same dataset as the one mentioned in this paper. Papadopoulos et al. [11] employed a variety of models, including CNN, LSTM, Bi-LSTM, GRU, and CAE, for classifying human activity recognition (HAR) using radar data. The CNN model outperformed the others, achieving a superior performance rate of 88%. Chakraborty et al. [12] utilized an X-band CW radar to compile the diverse DIAT-RadHAR dataset that captures activities such as army marching, stone pelting/grenade throwing, jumping while holding a gun, army jogging, army crawling, and boxing. This research also incorporated six pre-trained CNN models to support the deep learning (DL) architectures in analyzing the data, and the accuracy approached 98%. Zhu et al. [13] used CNN+LSTM for HAR and collected the data with Infineon’s Sense2GoL Doppler radar, achieving an accuracy of 98.28%. Noori et al. [14] used LSTM for the static actions with a non-wearable ultra-wideband (UWB) radar, with an accuracy approaching 99.6%. This study showed that DL architectures have better performance than machine learning models. Further, Li et al. [15] manually extracted human action features from both wearable and radar sensors, constructing a hybrid fusion model for enhanced action recognition.

Deep learning’s evolution has notably reduced the complexity associated with data preprocessing, enabling the use of more intricate and effective network models to boost recognition accuracy [16,17]. The research detailed in [18] employed position sensors alongside a multilayer perceptron (MLP), support vector machine (SVM), and Bayesian network (BN) for decision fusion, achieving superior recognition results. Further advancements have seen the micro-Doppler spectrogram transformed into a two-dimensional RGB image for recognition via 2D convolutional neural networks (2D-CNNs) [19,20]. With successes in natural language processing and speech recognition, the application of recurrent neural networks (RNNs) for temporally linked tasks has been deemed more data-appropriate [21]. Stacked long short-term memory (stacked LSTM) networks were proposed in [10] for time-frame-specific action recognition, while [17] demonstrated that bidirectional LSTM enhances recognition accuracy by simultaneously extracting time series features in both directions.

HAR poses a significant challenge in classification tasks, especially when similar actions yield comparable features, such as running and walking, in sensor-based tasks [22,23]. Thus, developing more effective preprocessing methods and model architectures remains a critical challenge. Radar sensing data, for instance, can produce spectra with distinct characteristics through various algorithms, with short-time Fourier transform (STFT), Hanning window (RIDHK), and smoothed pseudo-Wigner–Ville distribution (SPWVD) being some of the common techniques [24]. The VGG16 model has been explored for motion recognition, showcasing STFT’s superiority over other methods. The limitation of 2D-CNN in fully capturing human dynamics has led to proposals for architectures that blend 2D-CNN with 3D-CNN for better performance [25]. Another study, [26], emphasizes the integration of sensor data from multiple positions for HAR, introducing a deep learning framework that employs multiple channels. An advanced algorithm leverages principal component analysis (PCA) for coupling sensor data in the preprocessing phase. It then develops an innovative dual-channel convolutional neural network featuring an SPF (Spatial Pyramid Pooling) layer for effective integration [27]. Additionally, Le et al. [28] have applied the patching algorithm to STFT data for enhanced feature discrimination, utilizing sparse autoencoder (SAE) for gait recognition and leveraging the Bayesian optimization algorithm for optimal model parameter selection.

The conventional CNN model, while effective in extracting spatial features through local scopes, often neglects intra-layer feature interaction. To address this, Jin et al. [29] introduced the squeeze-and-excitation network (SENet), which employs global pooling and feature recalibration through channel-wise multiplication to enhance feature interaction within parallel convolutional layers, significantly improving feature discrimination. Building on the concept of Jin’s study, Huang et al. [30] suggested using an autoencoder (AE) for cross-channel communication in HAR tasks based on wearable sensors, marking a novel approach in strengthening CNNs’ spatial feature extraction capabilities for improved model performance in HAR applications. 

In the current HAR task, the model with the most generalization ability combines a CNN and an RNN to extract mixed features for recognition. RNNs, known for their numerous internal parameters, can be optimized for various data types and offer different variants for researchers. However, their capacity for temporal feature extraction has become saturated. Conversely, CNNs feature a relatively simple operational process and parameter settings, with limited literature suggesting improvements in spatial feature extraction capabilities. Therefore, this study aims to enhance the spatial feature extraction capabilities of CNNs to improve the model’s performance.

This study proposes constructing a HAR system based on a two-stream (2S) 1D CNN–BiGRU (bidirectional gated recurrent unit) model to recognize six daily activities: boxing, clapping, waving, walking, running, and standing still. Utilizing STFT to extract magnitude and phase parameters, the system inputs these to the 1D-CNN and BiGRU, respectively, for feature extraction and employs cross-channel operations (CCOs) in the 2S 1D-CNN. This process facilitates the exchange of magnitude and phase features in parallel convolutional layers, culminating in the merging of these features into the fully connected layer for model training and performance evaluation. The dataset, Rad-HAR [31], collected via radar sensors, consists of fast Fourier transform (FFT) data. Initially, the data are down-sampled and converted to STFT data for sample segmentation, with each segmented sample having a duration of 1 s. The 1D-CNN is utilized to extract spatial features, while the BiGRU extracts temporal features, and CCOs are implemented to exchange convolutional features, thereby maximizing the model’s performance.

## 2. Research Methods

The proposed HAR system, as depicted in Figure 1, is structured around three principal phases. Initially, the data preprocessing phase takes charge of eliminating the DC component from the dataset; transforming it into an STFT spectrum; and dividing the data into distinct sets for training (49%), validation (21%), and testing (30%). Following this, the model training phase involves introducing the training set to the 2S 1D-CNN+CCO-BiGRU model, which stands for 2S 1D-CNN combined with a cyclic cumulative output BiGRU. The final phase, ’Performance Evaluation’, assesses the model’s ability to classify inputs by applying the test set and using a confusion matrix to analyze the results.

### 2.1. Dataset

The dataset used in this study is the Rad-HAR public dataset [31] provided by the Temple University Advanced Signal Processing Laboratory, utilizing the Ancortek SDR-KIT 2500B (Ancortek Inc., Fairfax, VA, USA) [32] radar to collect information on eight types of user actions. Prior to the commencement of data collection activities, formal approval was secured from the Institutional Review Board (IRB) affiliated with Temple University, ensuring compliance with ethical standards and institutional guidelines. Figure 2 illustrates the 7.3 m × 3.5 m area monitored by Rad-HAR. Initially, a 25 GHz continuous wave is transmitted through a set of transmitters (TX) and, upon reflecting off an object, is captured by the receiver (RX) to collect data. The system stores FFT data at sampling frequencies of 512 kHz and 128 kHz. In each experiment, only one subject performed eight activities: boxing, hand clapping, hand waving, walking in place, running towards the radar, running away from the radar, walking towards the radar, and walking away from the radar. The model categorizes both running towards and away from the radar as ’running’ activity, and walking towards and away from the radar as ’walking’ activity, finally distinguishing six different activities.

### 2.2. Data Preprocessing

In the data preprocessing phase, we first used MATLAB to down-sample all data to 64 K samples per second. Data close to the DC component are irrelevant to the action and constitute clutter for the collected data. This clutter would negatively impact the model’s performance and must be removed. Next, we applied the STFT and selected the Hamming window with a 90% overlap to obtain the micro-Doppler spectrum. Figure 3 displays the STFT visualization for eight activities.

Then, we calculated the magnitude and phase of STFT. Here, $S_{r}\left(\omega \right)$ and $S_{i}\left(\omega \right)$ denote the real and imaginary parts of the STFT data, respectively, with *ω* representing the angular frequency. The magnitude and phase for time, *t*, and frequency, *f*, are represented by $A_{t}^{f}$ and $\phi_{t}^{f}$, respectively. The formulas for these calculations are provided in Equations (1) and (2). After the magnitude $A_{t}^{f}$ and phase $\phi_{t}^{f}$ are obtained, the data are stored in the form of a complex number, where the real part is the magnitude and the imaginary part is the phase, as shown in Equation (3).
(1)$$A_{t}^{f}=\sqrt{S_{r}^{2}(\omega )+S_{i}^{2}(\omega )}$$
(2)$$\phi_{t}^{f}=\tan^{-1}(S_{i}(\omega )/S_{r}(\omega ))$$
(3)$$x_{t}^{f}=A_{t}^{f}+j\phi_{t}^{f}$$

Data samples are then added using a sliding window. The original data cover a time span of 225 time frames (equivalent to 3 s). The original data are insufficient due to varying lengths of data for each activity. A sliding window technique can align these lengths, effectively augmenting the data. Through conducting experiments with overlap rates of 25%, 33.3%, and 50%, we found that a 33.3% overlap rate yields the best recognition performance. Thus, we segmented the data into samples of 75 time frames each (corresponding to 1 s), with each subsequent sample being shifted right by 50 time frames, resulting in an overlap of 25 time frames. As a result, we obtained 3216 samples for the training set and 1512 samples for the test set, as detailed in Table 1.

After observation, the distribution range of $A_{t}^{f}$ is 0 to 16,000, and the distribution range of $\phi_{t}^{f}$ is 0° to 360°. A significant disparity in the distribution ranges of $A_{t}^{f}$ and $\phi_{t}^{f}$ can impair the efficiency of model recognition. To mitigate this, we apply normalization, which scales the data to a range of 0 to 1 while maintaining the same ratio. The formula for normalization is provided in Equation (4), where X represents the sample data; $X_{max}$ and $X_{min}$ denote the maximum and minimum values of the data, respectively; and $X_{norm}$ is the normalized value.
(4)$$X_{norm}=\frac{X-X_{\min}}{X_{\max}-X_{\min}}$$

This study employed one-hot encoding to transform categorical labels into binary one-dimensional arrays, a crucial step in adapting these labels for computational models that require numerical input. Before conversion, each label represented a distinct activity, potentially signifying different attributes, conditions, or classifications relevant to our research. By applying one-hot encoding, we assigned each activity a unique binary vector, effectively eliminating any ordinal relationship and allowing the model to treat each activity with equal importance. Table 2 provides detailed binary vector conversions for each label.

### 2.3. Model Architecture

Figure 4 shows the model architecture proposed in this study, which includes three parts: The first part is the spatial feature extraction unit, composed of four layers of 2S 1D-CNN+CCO. The second part is the temporal feature extraction unit, consisting of two Bi-GRUs, with the magnitude (Mag) features and phase (Phase) features inputted into the Bi-GRUs. The third part consists of a concat layer and a three-layer fully connected (FC) layer, used for action recognition.

Figure 5 illustrates the 2S 1D-CNN+CCO architecture. The magnitude and phase features are separately input into the 2S 1D-CNN to extract the primary features (*Mag*_1_ and *Phase*_1_) and the secondary features (*Mag*_2_ and *Phase*_2_). The secondary features, after being multiplied by the weight parameter α, are passed to the *Real* and *Imag* components to integrate both the primary and secondary features. The method of calculation is presented in Equation (5).
(5)$$\begin{array}{l} Real=Mag_{1}\pm \alpha Phase_{2}\\ Imag=Phase_{1}\pm \alpha Mag_{2} \end{array}$$

In this study, the parameters for the 1D-CNN+CCO model were meticulously configured as follows: The size of the convolutional kernel was set to 3, and the architecture comprises convolutional layers with a descending number of layers, specifically 128, 64, 32, and 16 for each layer. The stride is configured at 1, the dropout rate at 0.3, and the weight parameter α at 1. The weight parameter α was set to 1 as a starting point for balancing different components of the loss function. This value of α was found to provide a stable training process and satisfactory results in initial tests, indicating a balanced emphasis on the model’s learning objectives. Regarding the activation function, the Rectified Linear Unit (ReLU) was selected for its efficacy in handling non-linear data transformations; ReLU operates by converting values below 0 to 0, while leaving values greater than 0 unchanged, as detailed in Equation (6).
(6)$$ReLU(x)=\left\{\begin{array}{l} x,\;if\;x>0\\ 0,\;if\;x\leq 0 \end{array}\right.$$

The parameters for the BiGRU are configured as follows: The number of GRU features is set to 128 for the one-way configuration and 256 for the two-way configuration. The FC layers contain 256, 128, and 64 neurons, respectively, with ReLU serving as the activation function. In this study, the model employs the Adam optimizer with an initial learning rate of 1 × 10^−4^. As the training process nears the global minimum, the relatively high learning rate may cause the model to overlook parameters associated with lower losses, thereby impeding the recognition of the optimal solution. To address this, the loss function is continuously monitored, and when the performance on the validation set plateaus for more than 10 epochs, the learning rate is decreased by 90% to facilitate the discovery of more favorable parameters.

The loss function employed in this study is categorical cross-entropy. The formula for its calculation is presented as follows in Equation (7):(7)$$loss=-\sum_{j=1}^{L}\sum_{i=1}^{C}q_{i,j}\log_{2}p_{i,j}$$

In the aforementioned formula, *L* denotes the total number of samples, and *C* represents the category. The term $q_{i,j}$ corresponds to the actual probability of the *i*-th category in the *j*-th data point, while $p_{i,j}$ signifies the predicted probability of the *i*-th category for the *j*-th data point by the model. A smaller loss value indicates superior model performance in terms of recognition.

### 2.4. Performance Evaluation

To evaluate the effectiveness of the model, this study employs a confusion matrix as a statistical tool to discern the discrepancies between predicted and actual outcomes. The analysis utilizes an untrained test set as input data. Figure 6 presents a schematic representation of the confusion matrix, which categorizes the model’s predictions into four key performance metrics: precision, recall, F1-score, and accuracy. These indicators are derived by comparing the predicted results against the actual ones, leading to four possible outcomes: True Positive (TP), False Positive (FP), True Negative (TN), and False Negative (FN). For illustration, consider the example of running:True Positive (TP): Actual running, predicted as running.False Positive (FP): Actual non-running, predicted as running.True Negative (TN): Actual non-running, predicted as non-running.False Negative (FN): Actual running, predicted as non-running.

The precision rate refers to the proportion of samples that the model accurately predicts as ‘running’ out of the total number of samples, as illustrated in Equation (8):(8)$$\mathrm{Precision}(\%)=\frac{TP}{TP+FP}\times 100\%$$

The recall rate is defined as the proportion of samples that are actually ‘running’ and are correctly recognized as ‘running’ by the model, as shown in Equation (9):(9)$$\mathrm{Recall}(\%)=\frac{TP}{TP+FN}\times 100\%$$

Relying only on precision or recall rates is not enough to assess a model’s excellence, which is why the F1-score is commonly used. This metric evaluates both indicators simultaneously, as shown in Equation (10). If either indicator is low, it significantly impacts the F1-score, making it a suitable measure for evaluating model performance.
(10)$$\mathrm{Accuracy}(\%)=\frac{TP+TN}{TP+TN+FP+FN}\times 100\%$$

## 3. Experimental Results

This study conducted experiments using a computer equipped with an Intel i7-10700 central processing unit (CPU), 32 GB of random access memory (RAM), and an NVIDIA GeForce RTX 2070 SUPER graphics processing unit (NVIDIA, Santa Clara, CA, USA). Anaconda 3 was used as the development environment for Python 3.7, along with the Anaconda 3 deep learning suite, which includes TensorFlow 2.1.0 and Keras 2.3.1. Jupyter Notebook was utilized to perform the training and performance evaluation of deep learning models.

### 3.1. Selection of the RNN Model

This study compares the recognition performance of various RNN models in capturing temporal features. The spatial feature extraction unit consists of four layers of 1D-CNN architecture, while the temporal feature extraction unit comprises a single layer of RNN models. The RNN models selected for this study included LSTM, GRU, bidirectional LSTM (BiLSTM), and bidirectional GRU (BiGRU), the numbers of which were 16, 16, 32, and 32, respectively.

Table 3 presents the performance of all models. The study results indicate that both the training and testing times for the bidirectional model are higher than those for the unidirectional model. However, the bidirectional model outperforms the unidirectional model in terms of recognition performance. Within the unidirectional models, GRU demonstrates better recognition performance than LSTM. Among the bidirectional models, BiLSTM slightly surpasses BiGRU in recognition performance. Nonetheless, due to its lower time cost, BiGRU has been selected for extracting temporal features in this study.

Table 4 presents the F1-scores obtained by using either magnitude or phase. We used magnitude and phase separately, trained the 1D-CNN-BiGRU model with each feature individually, and then evaluated the model’s recognition performance. When only magnitude is utilized, the F1-score reaches 92.1%, which is significantly higher than the 32.4% achieved when using only phase. However, when both magnitude and phase are used together, the F1-score can reach 96.3%.

### 3.2. Evaluating Cross-Channel Computing Layers

This study examines the effect of varying the number of application layers of 1D-CNN + CCO on recognition performance. The spatial feature extraction framework comprises four CNN layers, with the option to utilize either 1D-CNN or 1D-CNN + CCO for each layer to extract spatial features. The deployment of 1D-CNN + CCO is categorized into four scenarios based on its application across the layers:In the first scenario, a single layer of 1D-CNN + CCO is applied to one of the four layers (either layer 1, 2, 3, or 4), with the remaining three layers utilizing 1D-CNN. This configuration results in four distinct action recognition models.The second scenario involves using two layers of 1D-CNN + CCO, which can be applied to various combinations of layers (such as layers 1 and 2, 1 and 3, 1 and 4, 2 and 3, 2 and 4, or 3 and 4), leaving the other two layers to employ 1D-CNN. This approach yields six action recognition models.In the third scenario, three layers are equipped with 2S 1D-CNN+CCO, applied to combinations such as layers 1, 2, and 3; 1, 2, and 4; 1, 3, and 4; or 2, 3, and 4. The remaining layer uses 1D-CNN, also leading to six possible action recognition models.The fourth scenario explores the impact of utilizing 1D-CNN + CCO across all four layers, contrasting it with the performance when all four layers are configured with 1D-CNN.

Figure 7 displays the learning curves for the validation set of 2S 1D-CNN compared to the most effective combinations of 2S 1D-CNN + CCO. The findings of this study indicate that regardless of the number of 2S 1D-CNN + CCO layers implemented, both the accuracy and loss metrics for the validation set surpass those of the standalone 2S 1D-CNN. Furthermore, there is a clear trend showing that performance improves with an increase in the number of 2S 1D-CNN + CCO layers utilized. The optimum results are observed when all four layers are configured with the 2S 1D-CNN + CCO model.

In Table 5, we delineate the layer-wise model performance metrics, offering a comprehensive view of the incremental enhancements observed with the application of CCO. The empirical data drawn from this study provide a clear indication of the positive correlation between the number of CCO layers employed and the overall recognition performance of the model. Notably, each additional layer of CCO integrated into the architecture contributes to an accuracy increment in the range of 0.6 to 1%. This observation is particularly significant, as it suggests a linear relationship between the depth of CCO integration and the model’s accuracy in recognition tasks.

Focusing on a four-layered architecture, the empirical analysis reveals a distinct performance disparity between the standard 2S 1D-CNN model and its CCO-enhanced counterpart. The conventional four-layer 2S 1D-CNN shows an accuracy of 93.9%, which, while effective, is markedly surpassed by the 2S 1D-CNN+CCO configuration. The latter, incorporating CCO across all four layers, achieves a notably higher accuracy of 98.2%. This increase of 4.3% in accuracy is not merely statistically significant but also indicative of the substantive impact that CCO integration can have on the model’s efficacy.

### 3.3. Evaluating Cross-Channel Operation Parameters

This study explores the model recognition performance of 1D-CNN+CCO using different weight parameter α values and compares the main and auxiliary features using various feature merging methods. In this study, we adjust the weight parameters of the model to range from not utilizing auxiliary features (α = 0) to fully incorporating them (α = 1). This is achieved by setting α to 0, 0.2, 0.4, 0.6, 0.8, and 1.0, in order to train a total of six action recognition models. Table 6 presents the performance metrics for the models across the different α values. The experimental findings indicate that models with higher weight parameters, specifically at α = 1, demonstrate superior performance and achieve the best recognition capabilities.

This study also explored feature merging techniques within the real and imaginary components, employing either addition or subtraction to integrate the main and auxiliary features. This approach was used to train a total of four action recognition models. Table 7 provides a detailed analysis of the results obtained from various feature merging methods. The findings indicate that the technique of adding the two features together yields the most effective recognition performance. Overall, these results underscore the model’s strong adaptability and its ability to improve recognition accuracy by enhancing the spatial feature correlation between magnitude and phase.

### 3.4. Performance Comparison

In this study, we present an in-depth analysis of the activity recognition capabilities of three distinct computational models: 2S 1D-CNN+BiLSTM, 2S 1D-CNN+BiGRU, and our proposed 2S 1D-CNN+CCO+BiGRU. The evaluation spans various activities, denoted as B, HC, HW, P, R, and W, to encompass a broad spectrum of potential applications, as shown in Table 8. Figure 8 shows the activity-wise confusion matrix for all three models, (a) 2S 1D-CNN+BiLSTM, (b) 2S 1D-CNN+BiGRU, and (c) 2S 1D-CNN+CCO+BiGRU. 

The 2S 1D-CNN-BiLSTM model exhibited a robust performance, with precision rates ranging from 93.1% to 98.6% across the evaluated activities, leading to an overall average precision of 96.4%. The recall metrics closely followed this pattern, averaging 96.3%, and the F1-scores, which offer a balanced measure of precision and recall, mirrored this consistency with an average of 96.4%. The model’s overall accuracy was calculated at 96.4%, affirming its reliability in recognizing various activities. Furthermore, turning our attention to the 2S 1D-CNN-BiGRU model, we observed a comparable level of performance. Precision rates varied slightly but maintained an average of 96.4%. The recall metrics for this model also converged around a similar average, highlighting the model’s proficiency in accurately recognizing activities. This is further supported by the F1-scores and overall accuracy, both of which averaged 96.4%. The 2S 1D-CNN+CCO-BiGRU, however, marked a significant advancement in activity recognition performance. Precision rates peaked at 100% for certain activities, with an impressive average of 98.2% across the board. The recall rates were equally remarkable, averaging 98.2%, and the F1-scores aligned with these high standards, averaging 98.2% as well. The standout metric for this model was its overall accuracy, reaching 98.2%, which not only underscores its superior precision and recall capabilities but also highlights its potential as a highly effective tool for complex activity recognition tasks in various applications.

## 4. Discussion

This study marks a significant stride in the realm of activity recognition, a field pivotal to the burgeoning IoT landscape. By harnessing the capabilities of micro-Doppler radar data in conjunction with advanced deep learning techniques, this research not only amplifies the utility of IoT in daily life and emergency situations but also paves the way for novel approaches in monitoring and assistance technologies.

The innovative fusion of a 2S 1D-CNN with a BiGRU stands at the core of this study. This combination is adept at extracting and interpreting both spatial and temporal features from radar sensor data, processed initially through the STFT to derive time and frequency response information. The proposed CCO further refines the model’s efficacy by enhancing the interplay between magnitude and phase features across convolutional layers, a technique that showcases the potential for intricate feature manipulation within deep learning frameworks.

The employment of the Rad-HAR dataset for model training and validation underscores the practical applicability of the proposed system, with the model achieving a remarkable accuracy rate of 98.2%. This level of performance not only surpasses existing HAR systems but also highlights the robustness and reliability of deep learning when combined with micro-Doppler radar data for such applications.

This study also compares the computational efficiency of the models based on model parameters, floating point operations (FLOPs), and training and testing times. The 2S 1D-CNN+BiLSTM model requires 1.06 M parameters and 44.4 M FLOPs, with training and testing times recorded at 67.8 s and 1.85 s, respectively. The 2S 1D-CNN+BiGRU model shows a slight improvement in efficiency, utilizing 1.01 million parameters and the same number of FLOPs (44.4 M), while reducing the training and testing times to 64.9 s and 1.76 s, respectively. Our proposed model, the 2S 1D-CNN+CCO+BiGRU, demonstrates a significant reduction in the number of parameters to only 0.86 M, suggesting enhanced computational efficiency. However, this model also shows an increase in computational complexity, as indicated by 58.2 M FLOPs. As a result, the training time extends to 80.4 s, and the testing time per sample to 0.132 milliseconds (ms), as shown in Table 9. The length of the sample was 75 time frames, equivalent to 1 s. These factors are critically evaluated in the context of deploying deep learning models in edge computing environments for real-time recognition, where efficiency and the ability to operate with constrained resources are paramount. The proposed 2S 1D-CNN+CCO+BiGRU model, despite its increased computational complexity, offers a reduction in parameter size, potentially making it more suitable for real-time applications in edge computing scenarios.

This study also compares various methods with the proposed method, which utilizes a radar sensor and the same performance figures of merit used in compared studies, as shown in Table 10. The proposed method, 2S 1D-CNN+CCO+BiGRU, outperforms the other models in all metrics, achieving the highest precision, recall, F1-score, and accuracy, all at 98.2%. This indicates that the proposed model is exceptionally effective in accurately recognizing and classifying human activities. While other models such as DeepConvLSTM and VGG16 also exhibit high performance, the proposed model shows superior overall effectiveness. Conversely, DenseNet169 displays the lowest performance among the evaluated models. Although Ref. [13] also had an impressive accuracy rate of 98.3%, their data were collected by themselves. This comparative analysis highlights the strengths of the proposed 2S 1D-CNN+CCO+BiGRU model in the context of HAR, suggesting that its unique architecture and methodological advancements contribute to its high accuracy and efficiency in activity recognition tasks.

This research’s implications extend well beyond its immediate findings. The high accuracy and efficiency of the proposed 2S 1D-CNN+CCO+BiGRU model show the profound potential of integrating deep learning with micro-Doppler radar technology in creating sophisticated, non-intrusive HAR systems. Such systems could revolutionize the way we approach elder care, home automation, emergency response, and even workload reduction for caregivers, aligning closely with the ideals of improving quality of life and safety through technological innovation.

In this study, we used the Rad-HAR public dataset [31] to develop a deep learning model to recognize the activities. The resulting performances depended on the dataset, which was a limitation. However, this study opens avenues for future research, particularly in exploring the scalability of the proposed model across diverse environments and conditions and its adaptability to other forms of sensory data within the IoT framework. The exploration of cross-channel operations and their potential in enhancing data feature extraction could lead to even more sophisticated models, capable of nuanced activity recognition and interpretation. 

## 5. Conclusions

This study has successfully developed a HAR system using the 2S 1D-CNN+COO+BiGRU architecture. By converting radar-sensor-collected activity data (Rad-HAR) into STFT spectrum data and analyzing magnitude and phase information, the system leverages CNN and BiGRU for feature extraction. The integration of CCO facilitates the exchange of features across parallel convolutional layers, enabling the CNN to extract more discriminative spatial features, thereby enhancing recognition performance. The experimental outcomes were highly promising, demonstrating that the feature extraction capabilities of 2S 1D-CNN+CCO+BiGRU lead to superior activity recognition performance, with precision, recall, F1-score, and accuracy all achieving an impressive 98.2%. This performance surpasses that of existing activity recognition systems with a radar sensor, highlighting the significant application potential of the proposed model in the field of human activity recognition. This study has some limitations. We have employed the RAD-HAR open-source dataset which limits our control over the nature of parameters and categories. In the existing dataset, only one person was within the radar detection range. Therefore, we lack sufficient resources to verify the accuracy of recognition in a multi-person environment. This study also used this methodology with other Rad-HAR datasets. Moreover, since the data were recorded from a radar sensor that was placed at a fixed place, the limitation of this study is that the method cannot be used in an open field and can only be used for a single user, which can be addressed in future research.

## Figures and Tables

**Figure 1 sensors-24-02530-f001:**
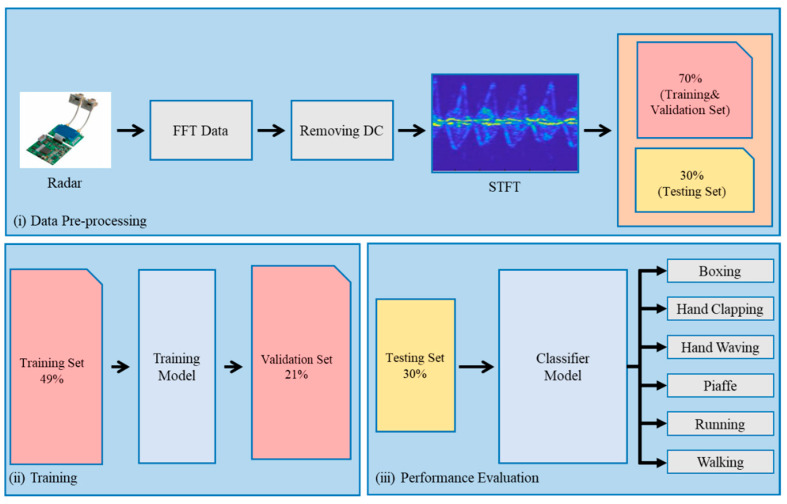
System architecture includes data preprocessing, model training, and testing results.

**Figure 2 sensors-24-02530-f002:**
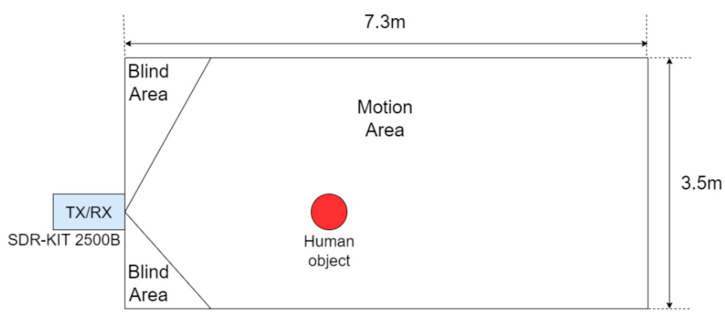
Rad-HAR dataset collection environment. There was one subject in the zone. Ancortek SDR-KIT 2500B radar was used to detect the activities.

**Figure 3 sensors-24-02530-f003:**
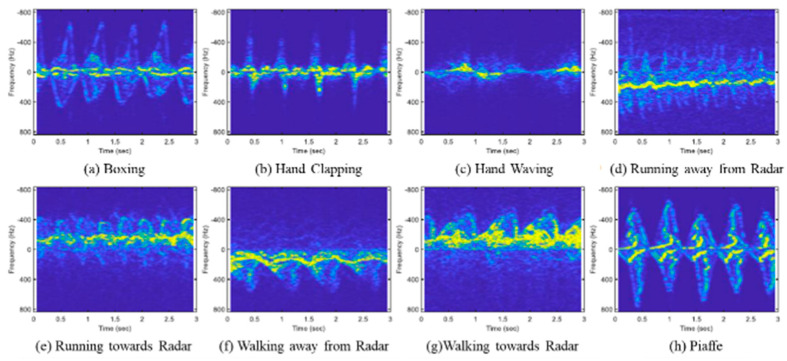
STFT visualization of eight activities.

**Figure 4 sensors-24-02530-f004:**
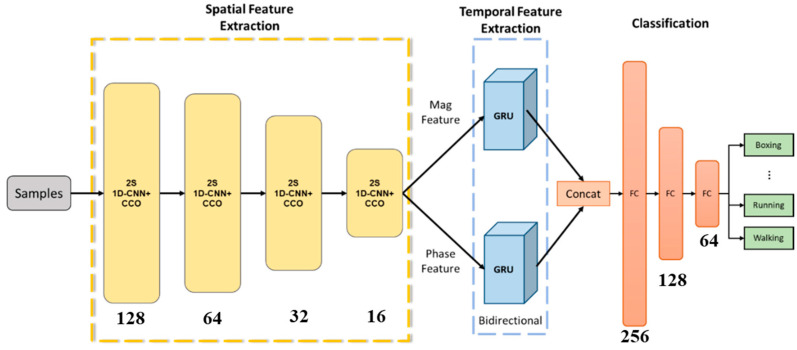
Model architecture.

**Figure 5 sensors-24-02530-f005:**
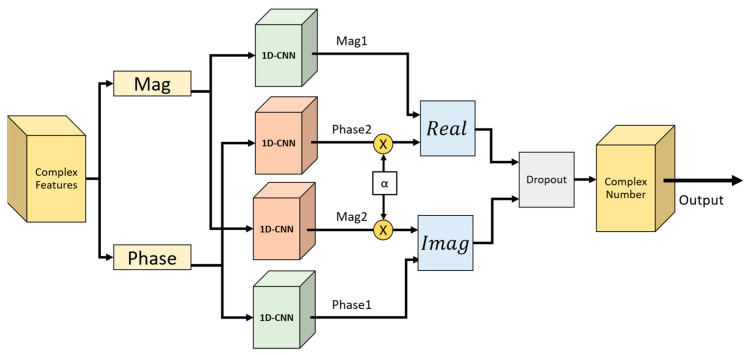
The 2S 1D-CNN+CCO architecture.

**Figure 6 sensors-24-02530-f006:**
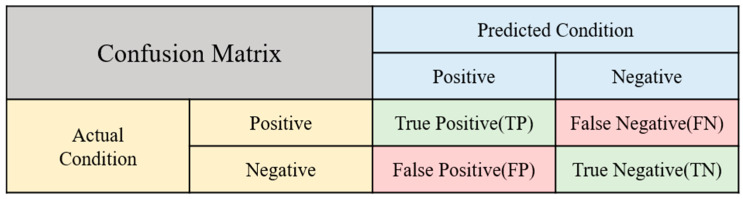
A schematic representation of the confusion matrix.

**Figure 7 sensors-24-02530-f007:**
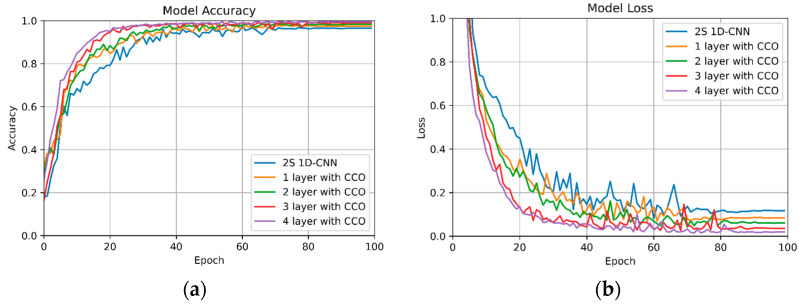
Learning curves of 2S 1D-CNN and best performance by layer with 2S 1D-CNN + CCO on the validation set: (**a**) validation set accuracy curve; (**b**) validation set loss curve.

**Figure 8 sensors-24-02530-f008:**
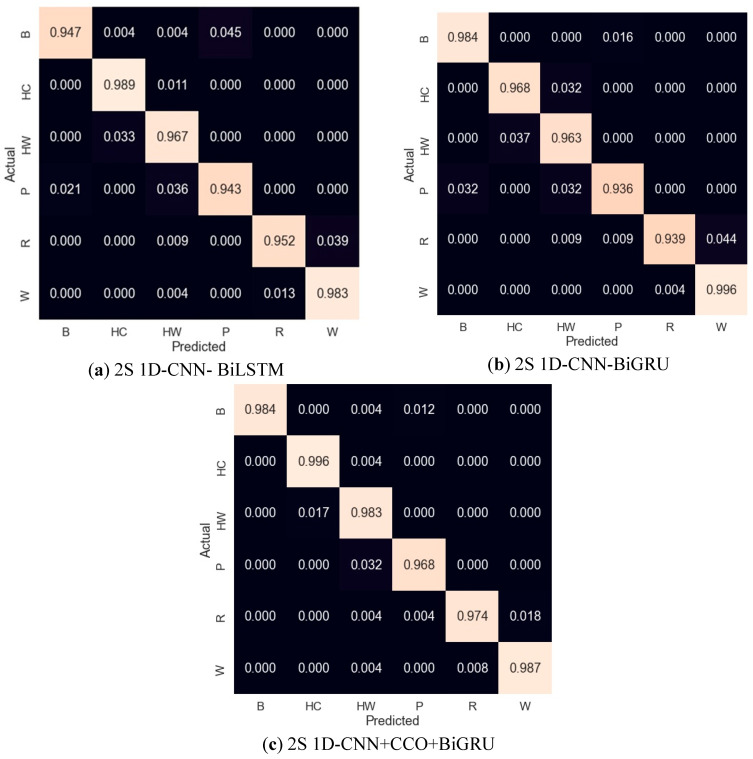
Confusion matrices of models: (**a**) 2S 1D-CNN- BiLSTM; (**b**) 2S 1D-CNN-BiGRU; (**c**) 2S 1D-CNN+CCO+BiGRU.

**Table 1 sensors-24-02530-t001:** Number of data preprocessing samples.

Activity	Raw Data Length (Seconds)		Training Samples	Test Samples
Boxing	591		544	244
Hand Clapping	597		512	284
Hand Waving	597		556	240
Walking in Place	597		516	280
Running+	190	570	532	228
Running−	380			
Walking+	391	594	556	236
Walking−	203			
Total	3546		3216	1512

Note: Running+ and Running− are running towards the radar and running away from the radar; Walking+ and Walking− are running towards the radar and walking away from the radar.

**Table 2 sensors-24-02530-t002:** Activity name code and one-hot code.

Activity Name	Code Name	One-Hot Code
Boxing	B	[ 1, 0, 0, 0, 0, 0]
Hand Clapping	HC	[ 0, 1, 0, 0, 0, 0]
Hand Waving	HW	[ 0, 0, 1, 0, 0, 0]
Walking in Place	P	[ 0, 0, 0, 1, 0, 0]
Running	R	[ 0, 0, 0, 0, 1, 0]
Walking	W	[ 0, 0, 0, 0, 0, 1]

**Table 3 sensors-24-02530-t003:** Recognition performance and computing efficiency of models.

Models	Precision	Recall	F1-Score	Accuracy	Training Time (sec)	Testing Time (ms/Sample)
LSTM	94.2	94.1	94.1	94.1	53.3	0.747
GRU	95.4	95.3	95.3	95.3	51.7	0.649
BiLSTM	96.5	96.3	96.4	96.4	67.8	1.223
BiGRU	96.4	96.3	96.3	96.3	64.9	1.167

**Table 4 sensors-24-02530-t004:** F1-score (%) of the model under different input characteristics.

Activity	Phase	Magnitude	Both
Boxing	25.9	90.4	97.4
Hand Clapping	21.7	93.3	96.8
Hand Waving	28.4	90.8	94.1
Walking in Place	39.9	91.6	95.6
Running	26.1	92.5	96.6
Walking	52.4	94.2	97.7
Total	32.4	92.1	96.3

**Table 5 sensors-24-02530-t005:** Layer-wise model performance metrics.

1st Layer	2nd Layer	3rd Layer	4th Layer	Precision (%)	Recall (%)	F1-Score (%)	Accuracy (%)	Parameters ^1^ (M)
✓	—	—	—	96.1	96.0	96.0	96.0	0.911
—	✓	—	—	95.7	95.7	95.7	95.7	0.985
—	—	✓	—	95.6	95.5	95.5	95.4	1.003
—	—	—	✓	95.4	95.3	95.3	95.4	0.985
Average				95.7	95.6	95.6	95.6	-
✓	✓	—	—	96.2	96.3	96.2	96.2	0.886
✓	—	✓	—	96.6	96.5	96.5	96.5	0.905
✓	—	—	✓	96.3	96.2	96.2	96.2	0.886
—	✓	✓	—	96.3	96.3	96.3	96.2	0.979
—	✓	—	✓	96.6	96.5	96.4	96.3	0.960
—	—	✓	✓	96.2	96.1	96.1	96.1	0.979
Average				96.4	96.3	96.3	96.2	-
✓	✓	✓	—	96.9	96.9	96.9	96.9	0.88
✓	✓	—	✓	97.1	97.1	97.1	97.1	0.862
✓	—	✓	✓	97.2	97.2	97.1	97.2	0.880
—	✓	✓	✓	97.2	97.2	97.2	97.1	0.954
Average				97.1	97.1	97.1	97.1	-
2S 1D-CNN				94.2	94.0	93.9	93.9	1.009
✓	✓	✓	✓	98.2	98.2	98.2	98.2	0.855

Parameters ^1^: The parameter is the size of the model.

**Table 6 sensors-24-02530-t006:** Comparative discrimination performance across different α values.

α	Precision (%)	Recall (%)	F1-Score (%)	Accuracy (%)
0	94.2	94.0	94.1	93.9
0.2	95.1	95.0	95.0	95.0
0.4	96.7	96.7	96.7	96.6
0.6	96.9	96.9	96.9	96.8
0.8	97.2	97.2	97.2	97.2
1	98.2	98.2	98.2	98.2

**Table 7 sensors-24-02530-t007:** Evaluating the recognition performance of 15 distinct feature merging methods.

CCO		Precision (%)	Recall (%)	F1-Score (%)	Accuracy (%)
Real	Imag				
−	−	94.4	94.4	94.2	94.1
−	+	97.8	97.8	97.8	97.8
+	−	96.4	96.3	96.2	96.2
+	+	98.2	98.2	98.2	98.2

**Table 8 sensors-24-02530-t008:** Comparison of activity recognition performance across three models.

Method	Performance	B	HC	HW	P	R	W	Average
2S 1D-CNN+BiLSTM	Precision (%)	94.5	96.9	93.1	96.0	98.6	96.2	96.4
	Recall (%)	94.7	98.9	96.7	94.3	95.2	98.3	96.3
	F1-score (%)	96.0	97.9	94.9	95.1	96.9	97.2	96.4
	Accuracy (%)	96.4						
2S 1D-CNN+BiGRU	Precision (%)	96.4	96.8	92.0	97.8	99.5	95.9	96.4
	Recall (%)	98.4	96.8	96.3	93.6	93.9	99.6	96.4
	F1-score (%)	97.4	96.8	94.1	95.6	96.6	97.7	96.4
	Accuracy (%)	96.4						
2S 1D-CNN+CCO+BiGRU	Precision (%)	100	98.6	94.8	98.5	99.1	98.3	98.2
	Recall (%)	98.4	99.6	98.3	96.8	97.4	98.7	98.2
	F1-score (%)	99.2	99.1	96.5	97.7	98.2	98.5	98.2
	Accuracy (%)	98.2						

**Table 9 sensors-24-02530-t009:** Comparing the computing efficiency of three models with all samples.

Method	Parameters (M)	FLOPs ^2^ (M)	Training Time (sec)	Testing Time (ms/Sample)
2S 1D-CNN-BiLSTM	1.06	44.4	67.8	0.122
2S 1D-CNN-BiGRU	1.01	44.4	64.9	0.116
2S 1D-CNN+CCO-BiGRU	0.86	58.2	80.4	0.132

FLOPs ^2^: Floating-point operations, used to measure the computational complexity of the model.

**Table 10 sensors-24-02530-t010:** Comparison of different methods with the proposed method that uses radar sensors.

Methods	Precision (%)	Recall (%)	F1-Score (%)	Accuracy (%)
DCNN [10]	83.0	83.0	83.0	83.0
LSTM [10]	96.9	93.3	93.1	93.0
CNN [11]	89.7	88.0	87.9	88.0
VGG16 [12]	96.8	96.7	96.7	97.0
1D-CNN+LSTM [13]	98	98	98	98.3
DenseNet169 [17]	68.4	68.3	68.3	69.2
**1D-CNN+CCO-BiGRU (Proposed)**	**98.2**	**98.2**	**98.2**	**98.2**

## Data Availability

Data are contained within the article.

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
