# Peer review of "Human Activity Recognition Based on Deep Learning and Micro-Doppler Radar Data"

_sensors, 2024, doi:10.3390/s24082530_

Round 1

Reviewer 1 Report

Comments and Suggestions for Authors

The topic of the paper is very interesting and findings in this area could be of great importance in practice. It is commendable that the authors have verified their results thoroughly and that they have provided extensive comparative analysis.

I have several questions:

-        For extraction of amplitude and phase parameters, STFT was used. However, STFT is the time-frequency distribution that does not have very good auto-terms concentration and can be unresilient to noise influence. How do you overcome this problem? Does your method work adequately in a noisy environment?

-        Eight types of user actions are observed, and very good results are achieved. Can this number be enlarged and how would this influence the execution time and real-time application of the method, keeping in mind that the method itself is complex?

-        In subsection 2.3. (lines 213-219), a set of parameters is defined. How do you determine their values? Please provide more info regarding parameters values selection. I.e. paramether alpha as a weight function is set to 1. Why?

Having in mind all the above I suggest a major revision of the paper.

Author Response

To Reviewer #1:

Thank the third reviewer for his/her valuable comments that make better this manuscript. The texts in this revised manuscript have been corrected/ modified by red words. It is our sincere hope that this revision will enhance readability and strengthen of the manuscript to satisfy the requirements of this prestigious journal.

Comments and Suggestions for Authors

The topic of the paper is very interesting and findings in this area could be of great importance in practice. It is commendable that the authors have verified their results thoroughly and that they have provided extensive comparative analysis.

I have several questions:

  1. For extraction of amplitude and phase parameters, STFT was used. However, STFT is the time-frequency distribution that does not have very good auto-terms concentration and can be unresilient to noise influence. How do you overcome this problem? Does your method work adequately in a noisy environment?

Response: Thank reviewer for yours comment. In this study, we used the open dataset [33] which  provided by the Temple University Advanced Signal Processing Laboratory, utilizing the Ancortek SDR-KIT 2500B [34] radar to collect information on eight types of user actions. When this dataset was collected, only one person was within the radar detection range. During the data pre-processing stage, we only removed background noise and did not include data involving multiple people in the detection range. Therefore, we lack sufficient resources to verify the accuracy of identification in a nosie enviroment. We add some sentences to mention the specifics of dataset more clear in section 2.1.

Lines 154-172

2.1. Dataset

The dataset used in this study is the Rad-HAR public dataset [33] provided by the Temple University Advanced Signal Processing Laboratory, utilizing the Ancortek SDR-KIT 2500B [34] radar to collect information on eight types of user actions. Prior to the commencement of data collection activities, formal approval was secured from the Institutional Review Board (IRB) affiliated with Temple University, ensuring compliance with ethical standards and institutional guidelines. Figure 2 illustrates the 7.3m × 3.5m area monitored by Rad-HAR. Initially, a 25 GHz continuous wave is transmitted through a set of transmitters (TX), which, upon reflecting off an object, is captured by the receiver (RX) to collect data. The system stores FFT data at sampling frequencies of 512 kHz and 128 kHz. In each experiment, only one subject performed the eight activities which included Boxing, Hand Clapping, Hand Waving, Piaffe, Running towards the Radar, Running away from the Radar, Walking towards the Radar, and Walking away from the Radar. The model categorizes both running towards and away from the Radar as 'running' activity, and walking towards and away from the Radar as 'walking' activity, finally distinguishing six different activities.

Figure 2. Rad-HAR dataset collection environment. There was one subject in the zone. Ancortek SDR-KIT 2500B radar was used to detect the activities.

  1. 2.Eight types of user actions are observed, and very good results are achieved. Can this number be enlarged and how would this influence the execution time and real-time application of the method, keeping in mind that the method itself is complex?.

Response: Thank reviewer for yours comment. While expanding the number of user actions observed by the method is feasible, the model would spend the much time in the training phase. Moreover, the accuracy would decrease. In Table 9, the total testing time is 2.01 seconds. Because the number of testing samples was 1512, the spending time for one sample was 0.132 ms. The testing time does not be changed when the number of samples increases. In order to decrease the misunderstanding, we modify the testing time with one sampl.

Lines 439-458

This study also compares the computational efficiency of the models based on model’s parameters, floating point operations (FLOPs), and training and testing times. The 2S 1D-CNN+BiLSTM model requires 1.06 M parameters and 44.4 M FLOPs, with training and testing times recorded at 67.8 seconds and 1.85 seconds, respectively. The 2S 1D-CNN+BiGRU model shows a slight improvement in efficiency, utilizing 1.01 million parameters and the same number of FLOPs (44.4 M), while reducing the training and testing times to 64.9 seconds and 1.76 seconds, respectively. Our proposed model, the 2S 1D-CNN+CCO+BiGRU, demonstrates a significant reduction in the number of parameters to only 0.86 M, suggesting enhanced computational efficiency. However, this model also shows an increase in computational complexity, as indicated by 58.2 M FLOPs. As a result, the training time extends to 80.4 seconds, and the testing time per sample to 0.132 millisecond (ms), as shown in Table 9. The length of sample was 75 time frames, equivalent to 1 second. These factors are critically evaluated in the context of deploying deep learning models in edge computing environments for the real-time recognition, where efficiency and the ability to operate with constrained resources are paramount. The proposed 2S 1D-CNN+CCO+BiGRU model, despite its increased computational complexity, offers a reduction in parameters size, potentially making it more suitable for real-time applications in edge computing scenarios.

Table 9. Comparing the computing efficiency of three models with all samples.

Method

Parameters(M)

FLOPs2(M)

Training Time(sec)

Testing Time (ms/sample)

2S 1D-CNN-BiLSTM

1.06

44.4

67.8

0.122

2S 1D-CNN-BiGRU

1.01

44.4

64.9

0.116

2S 1D-CNN+CCO-BiGRU

0.86

58.2

80.4

0.132

FLOP s 2: Floating -Point Operations, used to measure the computational complexity of the model.

  1. 3.In subsection 2.3. (lines 213-219), a set of parameters is defined. How do you determine their values? Please provide more info regarding parameters values selection. I.e. paramether alpha as a weight function is set to 1. Why?

Response: Thank reviewer for yours comment. In our study, the parameters for the 1D-CNN+CCO model were carefully configured based on a series of preliminary experiments and literature review (DCNN [10], LSTM [10], CNN [11], VGG16 [12], DenseNet169 [17] ) to optimize the model's performance. We add some sentences to mention more clear, “The weight parameter α was set to 1 as a starting point for balancing different components of the loss function. This value of α was found to provide a stable training process and satisfactory results in initial tests, indicating a balanced emphasis on the model's learning objectives.”. Then we used the grid search to find the optimal α and integration in Table 6 and Table 7.

Lines 237-247

In this study, the parameters for the 1D-CNN+CCO model were meticulously configured as follows: the size of the convolutional kernel was set to 3, and the architecture comprises convolutional layers with a descending number of layers, specifically 128, 64, 32, and 16 for each layer respectively. The stride is configured at 1, the Dropout rate at 0.3, and the weight parameter α at 1. The weight parameter α was set to 1 as a starting point for balancing different components of the loss function. This value of α was found to provide a stable training process and satisfactory results in initial tests, indicating a balanced emphasis on the model's learning objectives. Regarding the activation function, the Rectified Linear Unit (ReLU) was selected for its efficacy in handling non-linear data transformations; ReLU operates by converting values below 0 to 0, while leaving values greater than 0 unchanged, as detailed in Equation (6).

Lines 379-388

This study also explored feature merging techniques within the Real and Imaginary components, employing either addition or subtraction to integrate the main and auxiliary features. This approach was used to train a total of four action recognition models. Table 7 provides a detailed analysis of the results obtained from various feature merging methods. The findings indicate that the technique of adding the two features together yields the most effective recognition performance. Overall, these results underscore the model's strong adaptability and its ability to improve identification accuracy by enhancing the spatial feature correlation between amplitude and phase.

Table 6. Comparative discrimination performance across different α values.

α

Precision (%)

Recall(%)

F1-score(%)

Accuracy (%)

0

94.2

94.0

94.1

93.9

0.2

95.1

95.0

95.0

95.0

0.4

96.7

96.7

96.7

96.6

0.6

96.9

96.9

96.9

96.8

0.8

97.2

97.2

97.2

97.2

1

98.2

98.2

98.2

98.2

Table 7. Evaluating the recognition performance of 15 distinct feature merging methods.

CCO

Precision (%)

Recall(%)

F1-score(%)

Accuracy (%)

Real

Imag

-

-

94.4

94.4

94.2

94.1

-

+

97.8

97.8

97.8

97.8

+

-

96.4

96.3

96.2

96.2

+

+

98.2

98.2

98.2

98.2

Reviewer 2 Report

Comments and Suggestions for Authors

1.In 2.2 Data Pre-Processsing, you added data samples by using a sliding window, which yields an overlap rate of 33.3%. Does it affect the final outcome? Why or why not?

2.Compared to the model mentioned in this article, how is the performance of traditional machine learning models? (SVM, RF, KNN, etc.)

3.In Table 3, what data is used to train different models? What are the parameters?

4.Can the model achieve real-time recognition? If it can, how? If it cannot, how does it make sense?

5.What is the generalization ability of this model? Is it possible to directly use data from other radars?

Author Response

To Reviewer #2:

Thank the third reviewer for his/her valuable comments that make better this manuscript. The texts in this revised manuscript have been corrected/ modified by red words. It is our sincere hope that this revision will enhance readability and strengthen of the manuscript to satisfy the requirements of this prestigious journal.

Comments and Suggestions for Authors

1.In 2.2 Data Pre-Processsing, you added data samples by using a sliding window, which yields an overlap rate of 33.3%. Does it affect the final outcome? Why or why not?

Response: Thank reviewer for yours comment. Because the length of data for each activity was different, we used the sliding window technique to align these length, and effectively augment the samples. Through experimentation with overlapping rates of 25%, 33.3%, and 50%, we found that a 33.3% overlap rate yields the best model identification performance. We add some sentences to describe this process more clear.

Lines 192-200

Data samples are then added using a sliding window. The original data covers a time span of 225 time frames (equivalent to 3 seconds). The original data is insufficient due to varying lengths of data for each activity. A sliding window technique can align these lengths, effectively augmenting the data. Through conducting experiments with overlap rates of 25%, 33.3%, and 50%, we found that a 33.3% overlap rate yields the best recognition performance. Thus, we segmented the data into samples of 75 time frames each (corresponding to 1 second), with each subsequent sample being shifted right by 50 time frames, resulting in an overlap of 25 time frames. As a result, we obtained 3,216 samples for the training set and 1,512 samples for the test set, as detailed in Table 1.

2.Compared to the model mentioned in this article, how is the performance of traditional machine learning models? (SVM, RF, KNN, etc.)

Response: Thank reviewer for yours comment. This study employed deep learning models and also compared with various deep learning models. We didn’t compare our method with traditional machine learning models.

3.In Table 3, what data is used to train different models? What are the parameters?

Response: Thank reviewer for your comment. This study used 3,216 samples for the training set and 1,512 samples for the test set, as shown in Table 1. The RNN models selected for this study include LSTM, GRU, bidirectional LSTM (BiLSTM), and bidirectional GRU (BiGRU), which numbers were 16, 16, 32, and 32, respectively. We mention this information in Lines 298-204.

Lines 298-204

3.1. Selection of the RNN model

This study compares the recognition performance of various RNN models in capturing temporal features. The spatial feature extraction unit consists of four layers of 1D-CNN architecture, while the temporal feature extraction unit comprises a single layer of RNN models. The RNN models selected for this study include LSTM, GRU, bidirectional LSTM (BiLSTM), and bidirectional GRU (BiGRU), which numbers were 16, 16, 32, and 32, respectively.

4.Can the model achieve real-time recognition? If it can, how? If it cannot, how does it make sense?

Response: Thank reviewer for yours comment. Because the length of a sample was 75 time frames, corresponding to 1 second, and testing time of model was 0.132 ms, this model could approach the real-time recognition. We describe these conceptions clearer in discussion section.

Lines 439-456

This study also compares the computational efficiency of the models based on model’s parameters, floating point operations (FLOPs), and training and testing times. The 2S 1D-CNN+BiLSTM model requires 1.06 M parameters and 44.4 M FLOPs, with training and testing times recorded at 67.8 seconds and 1.85 seconds, respectively. The 2S 1D-CNN+BiGRU model shows a slight improvement in efficiency, utilizing 1.01 million parameters and the same number of FLOPs (44.4 M), while reducing the training and testing times to 64.9 seconds and 1.76 seconds, respectively. Our proposed model, the 2S 1D-CNN+CCO+BiGRU, demonstrates a significant reduction in the number of parameters to only 0.86 M, suggesting enhanced computational efficiency. However, this model also shows an increase in computational complexity, as indicated by 58.2 M FLOPs. As a result, the training time extends to 80.4 seconds, and the testing time per sample to 0.132 millisecond (ms), as shown in Table 9. The length of sample was 75 time frames, equivalent to 1 second. These factors are critically evaluated in the context of deploying deep learning models in edge computing environments for the real-time recognition, where efficiency and the ability to operate with constrained resources are paramount. The proposed 2S 1D-CNN+CCO+BiGRU model, despite its increased computational complexity, offers a reduction in parameters size, potentially making it more suitable for real-time applications in edge computing scenarios.

5.What is the generalization ability of this model? Is it possible to directly use data from other radars?

Response: Thank reviewer for yours comment. The model weights we propose cannot be directly applied to other radars data, as the parameters collected by different radars vary. This variation makes it difficult for the model to accurately identify data from other radars. However, by training through the model architecture we proposed, it is possible to achieve accurate recognition. Thus, we added sentences to describe this limitation in discussion section.

Lines 481-488

In this study, we used the Rad-HAR public dataset [33] to develop the deep learning model to recognize the activities. The performances of result depende

d on the dataset which was the limitation. However, the study opens avenues for future research, particularly in exploring the scalability of the proposed model across diverse environments and conditions, and its adaptability to other forms of sensory data within the IoT framework. The exploration of cross-channel operations and their potential in enhancing data feature extraction could lead to even more sophisticated models, capable of nuanced activity recognition and interpretation.

Reviewer 3 Report

Comments and Suggestions for Authors

The paper seems well written, the topic is relevant to the journal and the methods appear solid. however i have a few remarks:

1) The authors mention that the performance of their method (and in particular the accuracy) outperforms previous methods. This is a strong and important claim, yet i did not find a qualitative comparison anywhere in the manuscript. The only reference to this appears to me to be ref [9], where accuracy and other parameters of various methods are mentiond.

1.1 Is the comparison you are making fair?

1.2 Is it based on the same "Rad-HAR public dataset"?

1.3 Are your performance figures of merit counted in similar ways?

1.4 If yes, i think it could be of interest to the reader adding a table containing those figures for comparison. 

1.5 is there a trade-off for this increased accuracy? processing time as well as required hardware resources are important practical markers to determine just how useful your method can be (for example if it only runs fast enough given heavy GPU etc`, that will not be practical for many kinds of on-chip indoor sensors)

2. I worry that this is perhaps a very clean dataset. What happens if there is a runner AND a boxer in the scene? would your classifier work? what if there is yet another object outside the training set? like a passing car? how would that affect your method?

2.1 is your method perhaps too finely tuned for the particular sterile scattering scene? I suggest testing your trained model on a different 'clean' dataset taken elsewhere by another similar radar. would your performance metrics still be that good?

2.2 as mentioned before, you could easily synthesize a case of 2 or more activities carried out (simply by adding up the IQ maps from the radars), what would the performance look like than?

After answering those two questions I believe this could be an interesting publication  

Author Response

To Reviewer #3:

Thank the third reviewer for his/her valuable comments that make better this manuscript. The texts in this revised manuscript have been corrected/ modified by red words. It is our sincere hope that this revision will enhance readability and strengthen of the manuscript to satisfy the requirements of this prestigious journal.

Comments and Suggestions for Authors

The paper seems well written, the topic is relevant to the journal and the methods appear solid. however i have a few remarks:

1) The authors mention that the performance of their method (and in particular the accuracy) outperforms previous methods. This is a strong and important claim, yet i did not find a qualitative comparison anywhere in the manuscript. The only reference to this appears to me to be ref [9], where accuracy and other parameters of various methods are mentiond.

Response: Thank reviewer for your suggestion. We add one paragraph to describe the performances of existing studies with the redar sensors in introduction section.

Lines 65-84

In the previous studies, Wang et al. [10] combined stacked Recurrent Neural Networks (RNNs) with Long Short-Term Memory (LSTM) and deep Convolutional Neural Networks (CNNs) to classify six human actions—boxing, clapping, waving, piaffe, jogging, and walking—using radar Doppler images. Results showed the RNNs with LSTM achieved 92.65% accuracy, while deep CNNs reached 82.33%. The study utilized the same dataset as the one mentioned in this paper. Papadopoulos et al. [11] employed a variety of models, including CNN, LSTM, Bi-LSTM, GRU, and CAE, for classifying human activity recognition (HAR) using radar data. The CNN model outperformed the others, achieving a superior performance rate of 88%. Chakraborty et al. [12] utilized an X-band CW radar to compile the diverse DIAT-RadHAR dataset, which captures activities such as army marching, stone pelting/grenade throwing, jumping while holding a gun, army jogging, army crawling, and boxing. This research also incorporated six pre-trained CNN models to support the deep learning (DL) architectures in analyzing the data, and the accuracy approached to 98%. Zhu et al. [13] used the CNN+LSTM for HAR, and collected the data with Infineon’s Sense2GoL Doppler radar, achieving the accuracy of 98.28%. Noori et al. [14] used the LSTM for HAR with a non-wearable ultra-wideband (UWB) radar, aiming to efficiently process and learn from time-series data typical of activity recognition tasks. Further, Li et al. [15] has manually extracted human action features from both wearable and radar sensors, constructing a hybrid fusion model for enhanced action recognition.

1.1 Is the comparison you are making fair?

Response: Thank reviewer for your comment. In Table 10, we compared the proposed model with the other methods with the radar sensors. Where DCNN [10] used the Rad-HAR public dataset. We modify this mention.

Lines 460-473

This study also compares various methods with the proposed method, which utilizes radar sensor and same performance figures of merit used in compared studies, as shown in Table 10. The proposed method, 2S 1D-CNN+CCO+BiGRU, outperforms the other models in all metrics, achieving the highest precision, recall, f1-score, and accuracy, all at 98.2%. This indicates that the proposed model is exceptionally effective in accurately identifying and classifying human activities. While other models such as DeepConvLSTM, and VGG16 also exhibit high performance, the proposed model shows superior overall effectiveness. Conversely, DenseNet169 displays the lowest performance among the evaluated models. Although, Ref [13] also had an impressive accuracy rate of 98.3%, their data were collected by themselves. This comparative analysis highlights the strengths of the proposed 2S 1D-CNN+CCO+BiGRU model in the context of HAR, suggesting that its unique architecture and methodological advancements contribute to its high accuracy and efficiency in activity recognition tasks.

Table 10. Comparison of different methods with the proposed method that uses radar sensors.

Methods

Precision(%)

Recall(%)

F1-score(%)

Accuracy(%)

DCNN [10]

83.0

83.0

83.0

83.0

LSTM [10]

96.9

93.3

93.1

93.0

CNN [11]

89.7

88.0

87.9

88.0

VGG16 [12]

96.8

96.7

96.7

97.0

1D-CNN+LSTM [13]

98

98

98

98.3

DenseNet169 [17]

68.4

68.3

68.3

69.2

1D-CNN+CCO-BiGRU(Proposed)

98.2

98.2

98.2

98.2

1.2 Is it based on the same "Rad-HAR public dataset"?

Response: Thank reviewer for your comment. We modify the mention.

1.3 Are your performance figures of merit counted in similar ways?

Response: Thank reviewer for your suggestion. In this study, we compared our proposed model with previous studies (DCNN [10], LSTM [10], CNN [11], VGG16 [12], 1D-CNN+LSTM [13], and DenseNet169 [17]). Where we used same performance figures of merit in Table 10.

1.4 If yes, i think it could be of interest to the reader adding a table containing those figures for comparison. 

Response: Thank reviewer for your suggestion. We have addressed this mention in this paragraph.

1.5 is there a trade-off for this increased accuracy? processing time as well as required hardware resources are important practical markers to determine just how useful your method can be (for example if it only runs fast enough given heavy GPU etc`, that will not be practical for many kinds of on-chip indoor sensors)

Response: Thank reviewer for your comment. We describe the specifics of used hardware in results section. Moreover, because the length of a sample was 75 time frames, corresponding to 1 second, testing time of model was 0.132 ms, and the number of model’s parameters was 0.85 M, this model could be practiced in the edge computing for the real-time recognition. We describe the information in discussion section.

Lines 291-297.

  1. Experimental Results

This study conducted experiments using a computer equipped with an Intel i7-10700 Central Processing Unit (CPU), 32 GB of Random Access Memory (RAM), and an NVIDIA GeForce RTX 2070 SUPER Graphics Processing Unit (GPU). Anaconda 3 was used as the development environment for Python 3.7, along with the Anaconda 3 deep learning suite, which includes TensorFlow 2.1.0 and Keras 2.3.1. Jupyter Notebook was utilized to perform the training and performance evaluation of deep learning models.

Lines 439-457

This study also compares the computational efficiency of the models based on model’s parameters, floating point operations (FLOPs), and training and testing times. The 2S 1D-CNN+BiLSTM model requires 1.06 M parameters and 44.4 M FLOPs, with training and testing times recorded at 67.8 seconds and 1.85 seconds, respectively. The 2S 1D-CNN+BiGRU model shows a slight improvement in efficiency, utilizing 1.01 million parameters and the same number of FLOPs (44.4 M), while reducing the training and testing times to 64.9 seconds and 1.76 seconds, respectively. Our proposed model, the 2S 1D-CNN+CCO+BiGRU, demonstrates a significant reduction in the number of parameters to only 0.86 M, suggesting enhanced computational efficiency. However, this model also shows an increase in computational complexity, as indicated by 58.2 M FLOPs. As a result, the training time extends to 80.4 seconds, and the testing time per sample to 0.132 millisecond (ms), as shown in Table 9. The length of sample was 75 time frames, equivalent to 1 second. These factors are critically evaluated in the context of deploying deep learning models in edge computing environments for the real-time recognition, where efficiency and the ability to operate with constrained resources are paramount. The proposed 2S 1D-CNN+CCO+BiGRU model, despite its increased computational complexity, offers a reduction in parameters size, potentially making it more suitable for real-time applications in edge computing scenarios.

Table 9. Comparing the computing efficiency of three models with all samples.

Method

Parameters(M)

FLOPs2(M)

Training Time(sec)

Testing Time (ms/sample)

2S 1D-CNN-BiLSTM

1.06

44.4

67.8

0.122

2S 1D-CNN-BiGRU

1.01

44.4

64.9

0.116

2S 1D-CNN+CCO-BiGRU

0.86

58.2

80.4

0.132

FLOP s 2: Floating -Point Operations, used to measure the computational complexity of the model.

  1. I worry that this is perhaps a very clean dataset. What happens if there is a runner AND a boxer in the scene? would your classifier work? what if there is yet another object outside the training set? like a passing car? how would that affect your method?

Response: Thank reviewer for your comment.  We have employed the RAD-HAR open-source dataset which limits our control over the nature of parameters and categories. In the existing dataset, only one person was within the radar detection range. During the data pre-processing stage, we only removed background noise and did not include data involving multiple people in the detection range. Therefore, we lack sufficient resources to verify the accuracy of identification in a multi-person environment. We will address this limitation in future research.

2.1 is your method perhaps too finely tuned for the particular sterile scattering scene? I suggest testing your trained model on a different 'clean' dataset taken elsewhere by another similar radar. would your performance metrics still be that good?

2.2 as mentioned before, you could easily synthesize a case of 2 or more activities carried out (simply by adding up the IQ maps from the radars), what would the performance look like than?

Response for 2.1 and 2.2:  Thank you for your suggestion. The performances of proposed model depended on the data. We understand that performances will decrease under the background-noise enviroment. The previous studies also performed the HAR with the clear data We will consider this as the limitation of our study and proceeed it in the future study. We add some sentences in conclusion section.

Lines 489-508

  1. Conclusions

This study has successfully developed a HAR system using the 2S 1D-CNN+COO+BiGRU architecture. By converting radar sensor-collected activity data (Rad-HAR) into STFT spectrum data and analyzing vibration and phase information, the system leverages CNN and BiGRU for feature extraction. The integration of CCO facilitates the exchange of features across parallel convolutional layers, enabling the CNN to extract more discriminative spatial features, thereby enhancing recognition performance. The experimental outcomes have been highly promising, demonstrating that the feature extraction capabilities of 2S 1D-CNN+CCO+BiGRU lead to superior activity recognition performance, with precision, recall, F1-score, and accuracy all achieving an impressive 98.2%. This performance surpasses that of existing activity recognition systems with the radar sensor, highlighting the significant application potential of the proposed model in the field of human activity recognition. This study has some limitations, we have employed the RAD-HAR open-source dataset which limits our control over the nature of parameters and categories. In the existing dataset, only one person was within the radar detection range. Therefore, we lack sufficient resources to verify the accuracy of identification in a multi-person environment. This study also used this methodology with others Rad-HAR dataset. Moreover, since the data is recorded from the radar sensor that was placed at a fixed place, the limitation of this study is that the method could not be used at an open field, and only for the single user, which can be addressed in future research.

After answering those two questions I believe this could be an interesting publication  

Round 2

Reviewer 1 Report

Comments and Suggestions for Authors

All my concerns from the previous review round have been resolved. I believe that this paper will be useful to the future readers. I suggest acceptance of the paper.

Reviewer 3 Report

Comments and Suggestions for Authors

i am satisfied with the author's response